# The Role of *Momordica charantia* in Resisting Obesity

**DOI:** 10.3390/ijerph16183251

**Published:** 2019-09-04

**Authors:** Meiqi Fan, Eun-Kyung Kim, Young-Jin Choi, Yujiao Tang, Sang-Ho Moon

**Affiliations:** 1Division of Food Bioscience, College of Biomedical and Health Sciences, Konkuk University, Chungju 27478, Korea; 2School of Bio-Science and Food Engineering, Changchun University of Science and Technology, Changchun 130–600, China

**Keywords:** *Momordica charantia*, bioactive components, anti-obesity, mechanisms

## Abstract

*Momordica charantia* (*M. charantia*), commonly known as bitter gourd, bitter melon, kugua, balsam pear, or karela, is a tropical and sub-tropical vine belonging to the Cucurbitaceae family. It has been used to treat a variety of diseases in the traditional medicine of China, India, and Sri Lanka. Here, we review the anti-obesity effects of various bioactive components of *M. charantia* established at the cellular and organismal level. We aim to provide links between various bioactive components of *M. charantia* and their anti-obesity mechanism. An advanced search was conducted on the worldwide accepted scientific databases via electronic search (Google Scholar, Web of Science, ScienceDirect, ACS Publications, PubMed, Wiley Online Library, SciFinder, CNKI) database with the query TS = “*Momordica charantia*” and “obesity”. Information was also obtained from International Plant Names Index, Chinese Pharmacopoeia, Chinese herbal classic books, online databases, PhD and MSc dissertations, etc. First, studies showing the anti-obesity effects of *M. charantia* on the cells and on animals were classified. The major bioactive components that showed anti-obesity activities included proteins, triterpenoids, saponins, phenolics, and conjugated linolenic acids. Their mechanisms included inhibition of fat synthesis, promotion of glucose utilization, and stimulation of auxiliary lipid-lowering activity. Finally, we summarized the risks of excessive consumption of *M. charantia* and the application. Although further research is necessary to explore various issues, this review establishes the therapeutic potential of *M. charantia* and it is highly promising candidate for the development of anti-obesity health products and medicines.

## 1. Introduction

According to World Health Organization (WHO), the incidence of obesity has increased by more than twofold since 1980. In June 2013, the American Medical Association officially regarded obesity as a disease [1]. Meanwhile, in 2014, more than 1.9 billion adults were overweight, including 600 million obese people [2,3]. Moreover, obesity is one of the major risk factors for the development of chronic diseases, such as hypertension, dyslipidemia, diabetes mellitus type 2, cardiovascular disease, insomnia, and apnea [4]. Therefore, obesity has become a major concern for the entire society. Although obesity can be treated with drugs and surgery, they may cause severe side effects in the body [5].

In recent years, increased attention is being paid to the development of novel therapies for the treatment of obesity [6,7]. Changes in lifestyle and restrictions on fat intake are considered to be the useful to manage obesity. However, owing to the accelerated pace of modern life, increased stress, and the concerns regarding the safety and tolerability of anti-obesity drugs, anti-obese effects of functional food are being explored. Currently, the development and utilization of plant resources are actively pursued in the field of functional food. Therefore, research on natural products and their anti-obesity activities has become a major topic of interest in the recent years.

*Momordica charantia* (*M. charantia*), commonly known as bitter gourd, bitter melon, kugua, balsam pear, or karela, is a climber belonging to the Cucurbitaceae family. It bears oblong fruits resembling cucumber, and the tender fruits are emerald green in color and turn to orange-yellow on ripening [8,9,10]. As a medicinal and edible vegetable, *M. charantia* has been used in the traditional medicine of China, India, and Sri Lanka for a long time to treat a variety of diseases [11,12,13]. There are different varieties of *M. charantia* with different growing environments and different growing sites [14]. In traditional Chinese treatments, *M. charantia* fruits, seeds, stem, and leaves are used as medicine for intestinal diseases, hypoglycemia, cancer, and some of viral infections. In India, Sri Lanka, and other countries, *M. charantia* is used to prepare nutritious food, hypoglycemic drugs, emetics, and laxative drugs [15]. Modern medical research shows that *M. charantia* exhibits anti-tumor, hypoglycemic, hypolipidemic, anti-oxidant, anti-viral, anti-bacterial, immune regulation, and other effects [16]. The active ingredients isolated from *M. charantia* include triterpenes, steroids, polysaccharides, proteins, peptides, and alkaloids [17]. Lakholia, a physician, was probably the first to document the therapeutic effect of *M. charantia* in 1956, using himself as the subject [18]. Subsequently, several studies have shown that *M. charantia* extract has anti-obesity activity, thus, exhibits high clinical value [19,20,21]. However, only a few review articles are available on this subject. The accumulating evidence indicates some papers show contradictory results. This has to do with the complex metabolic mechanisms of obesity. Also, this makes it difficult to elucidate the anti-obesity effect of *M. charantia*.

In this review, we discuss the inhibitory activity of *M. charantia* on the accumulation of fat, in addition to the major mechanisms of the corresponding bioactive substances. Our aim was to summarize the literature findings that provide the fundamental ideas for efficient processing and utilization of *M. charantia* for anti-obesity treatment.

## 2. Anti-Obesity Components of *M. charantia*

### 2.1. Saponins and Triterpenes

Saponins are plant secondary metabolites that occur in the roots, stems, leaves, seeds, and other plant tissues of a wide range of plant sources [22]. Saponins are classified into two broad categories, namely, steroidal saponins and triterpenoid saponins [23]. Sugimoto et al., found that triterpenes extracted from the leaves of Hollyhock using methanol, can inhibit the hydrolytic activity of pancreatic lipase to achieve the anti-obesity effect [24]. IC_50_ value of the triterpenoid saponin extract that could inhibit pancreatic lipase hydrolysis activity was 3.63 mg/mL [25]. Zhao et al., found that the Campanulaceae saponins rich diet can reduce the body weight of mice on high-fat diet [26]. As early as 1986, it was found that saponins in *M. charantia* seeds ameliorated obesity by improving insulin sensitivity and glucose homeostasis. [27].

The *M. charantia* roots, stems, leaves, and fruits contain saponins, mainly triterpenoid saponins. Experiments have shown that the total content of saponins in *M. charantia* seeds is approximately 0.432% [28]. Many types of saponins have been isolated from *M. charantia*, including cucurbit, oleanane-type, ursulan type, stigmasterol, cholesterol, and sitosterol saponins [29,30,31]. These ingredients have been found to reduce visceral fat weight and glucose levels, promote oxidation in liver and adipose tissue, and significantly lower blood TG levels [32,33,34,35]. In 1980, subsequent to the extraction of bitter ginseng, Okabe H isolated broad bean glycosides, *M. charantia* saponin-A and -B [36]. In 1981, Miyahara Y. et al. isolated saponin-C, -D, and -E from *M. charantia* extract [37]. In 1982, Okabe H. et al., isolated saponins-G, -F1, -F2, -I, -K, and -L from the extract of immature fruit of *M. charantia* [38,39]. Meanwhile, a saponin-rich fraction isolated from *M. charantia* was also found to lower blood sugar and small intestine disaccharidase activity in rats [40]. In 1984, Yasuda M. et al., isolated momordicine-I, -II, and -III from stem and leaf extracts of *M. charantia* [41]. They also isolated *M. charantia* glycoside-A and -B [42]. In 1995, Chang et al., isolated four compounds from the *M. charantia* fruit extract. Among these, three compounds were tetracyclic triterpenoid fennel compounds (*M. charantia* glycosides -F1, -G, and -I) [43]. In 1997, Begum S. et al., used the fresh extract *M. charantia* to obtain momordicine, momordicinin, and momordicilin [44]. In 2001, Toshiyuki M. et al., used the extract of *M. charantia* and isolated goyaglycoside -a, -b, -c, -d, -e, -f, -g, and -h as well as goyasaponin -I, -II, and -III [45]. In 2010, a new anti-diabetic formulation with *M. charantia* active ingredients, such as momordicines and glycosides, was developed [46]. In 2005, Tian et al., crushed fresh *M. charantia*, separated the extract and identified momordicoside-A [47]. In 2007, Pan et al., isolated three compounds from the dry and immature fruits of *M. charantia*, namely, germanicyl acetate, aglycone of momordicoside-I and -L [48]. In the same year, Guan et al. used the immature fruit extract of *M. charantia* and 95% ethanol refluxing, to obtain aglycones, such as momordicoside-F1 and -I [49]. Momordicosides have been shown to enhance fatty acid oxidation in in vivo models [50]. Studies have also shown that saponins inhibit pancreatic lipase activity and are considered to have anti-obesity and cholesterol-lowering properties [51]. In 2008, Cheng et al., isolated 26, 27-dihydroxy lanolin-7, 9 (11), 24-triene(24R)-cyclohexanone-3α, 24R, 25-triol, which is saponin, from the extract of *M. charantia* stem and leaf [52].

Saponins and triterpenes isolated from *M. charantia*, are presented in Figure 1.

### 2.2. Phenolics and Flavonoids

The phenolic compounds isolated from *M. charantia* were quinic acid, protocatechuic acid, ascorbic acid, gallic acid, chlorogenic acid, catechin, syringic acid, caffeic acid, rutin, 4-coumaric acid, myricetin, gentisic acid, vanillic acid, coumaric acid, t-cinnamic acid, and p-methoxy-benzoic acid. These have been found to exhibit anti-oxidant activity [53,54,55]. The flavonoid compounds have a polyphenol structure which is present in *M. charantia* in 2 forms, namely, free-aglycones and glycosides that bind to sugars [56]. The concentration of flavonoids in the ethanol extract and the water extract of *M. charantia* was 44.0 and 62.0 mg/g, respectively, while the concentration of polyphenols was 68.8 and 51.6 mg/g, respectively [57]. However, there is no clear data on the anti-obesity activity of phenols and flavonoids in *M. charantia*.

Meanwhile, the potential anti-obesity effects of phenolic compounds in animals have been reported in several studies. Chen et al. oral administration of a dose of polyphenol (100 mg/kg BW) to mice for 30 days and found that the body weight of mice and serum TC levels were significantly reduced compared with that of the control group [58]. Further, several flavonoids have been shown to play an important role in the treatment of obesity. In one of the studies, the catechins (5.7 g/kg BW) were oral administered to mice for 4 weeks. It was observed that catechins can decrease the body weight, levels of serum biochemical indicators (TC and TG), and levels of liver lipid [59]. In summary, several in vivo experimental results showed that the administration of phenolic compounds exhibited similar effect as that of lipid-lowering diet on mice. Although, *M. charantia* contains lots of phenols and flavonoids mentioned above, the bioactivity of phenols and flavonoids in *M. charantia* is still curious. This provides a possibility for the future research of phenols and flavonoids in *M. charantia*.

Figure 2 illustrates the phenolic constituents isolated from *M. charantia* using high performance liquid chromatography (HPLC).

### 2.3. Unsaturated Fatty Acids

As the unsaturated fatty acids constitute a major component of the body fat, their impact on lipid metabolism is a topic of interest worldwide. The unsaturated fatty acids inhibit the activity of fatty acid synthase (FAS), diglyceride transacylase (DGAT), and hydroxymethyl glutaryl coenzyme A (HMG-CoA) reductase. They promote the oxidative decomposition of fatty acids to inhibit glycerol TG synthesis, to down-regulate low-density lipoprotein (LDL) receptors in the liver, and to inhibit total cholesterol (TC) synthesis and absorption, thus, reducing serum TG and TC content [60,61]. Takada et al. showed that unsaturated fatty acids could stimulate rat liver peroxidase and can participate in the induction of uncoupling protein (UCP) and enhance β-oxidation in mitochondria in the rat liver, thereby accelerating the breakdown and clearance of lipids in the liver, and subsequently reducing the body fat deposition [62]. Baillie et al., found that dietary supplementation with fish oil induces peroxisomal fatty acid oxidation of skeletal muscle and increases UCP3 levels by a factor of 2, thereby reducing fat and protein deposition [63].

The proportion of unsaturated fatty acids in the *M. charantia* is high. The ratio of monounsaturated fatty acid content to total fatty acid content is approximately 20.1%, and the proportion of polyunsaturated fatty acid is approximately 64.3% (Figure 3). At present, nine kinds of unsaturated fatty acids have been extracted from *M. charantia* [56]. *M. charantia* seed oil is a unique as it contains 9cis, 11trans, 13trans-conjugated linolenic acid (9c, 11t, 13t-CLN) at high levels [64]. Ahmad et al., have tested α-glucosidase and α-amylase inhibitory activity and showed unusually high amounts of C18-0 (stearic acid, 62.31% of total fatty acid). C18-1 (oleic acid) and C18-2 (linoleic acid) were the other major fatty acid detected (12.53% and 10.40%, respectively). Other fatty acids detected in the analysis included C10-0 (0.53%, capric acid), C12-0 (0.45%, lauric acid), C16-0 (4.72%, palmitic acid) and C20-0 (5.3%, arachidic acid) [65]. In 2001, Noguchi et al. studied the effects of dietary fat consisted of soybean oil + 2.0% *M. charantia* seed oil on blood and liver fat metabolism and found that the intake of *M. charantia* seed extract can increase the content of linoleic acid (c-9, t-11-CLA) in the liver and in turn, decreases the free fatty acids in the blood [66]. In 2011, Gadang et al. found that fed AIN-93M diet modified to contain 3.0% (*wt*/*wt*) ground *M. charantia* seed for 100 days can promote the expression of peroxisome proliferator-activated receptor Y gene in the adipose tissue, reduce the expression of nuclear factor kappa B and further alleviate the symptoms associated with metabolic syndromes [67]. In 2012, a study by Chan et al. showed that containing 15% *M. charantia* extract diet reduces accumulation of body fat by regulating protein kinase activity and programmed cell death in adipocytes [68]. Moreover, containing 20% *M. charantia* seed extract diet extract exhibits the weight loss should be due to white adipose tissue degradation, inflammation and browning effect [69]. In 2017, Chen et al. found that the anti-adiposity function of *M. charantia* seed oil was solely attributed to its fatty acid fraction and that the FFA form was more effective than the TG form [70].

## 3. Anti-Obesity Mechanism of *M. charantia*

Anti-obesity mechanism of *M. charantia* was summarized in the Table 1.

### 3.1. Inhibition of the Synthesis of Fat

Fat synthesis is considered to be a key step in the development of obesity and regulating this step would effectively control the progression of obesity [89]. Currently, 3T3-L1 adipocytes are the important models used globally for studying the molecular mechanism of adipogenesis. The data showed that the 50% Ethanolic Extract of *M. charantia* is potent inhibitor of lipogenesis and stimulator of lipolysis in 3T3-L1 pre-adipocytes [90]. In 3T3-L1 cell differentiation studies, it has been found that adipogenesis is a highly elaborate regulatory process that involves a variety of transcription factors, such as C/EBPα and PPARγ) and lipid metabotropic proteins, such as, GluT-4, adiponectin, and leptin) [91,92].

Tan et al. showed that treatment of L6 myotubes and 3T3-L1 adipocytes with the extract of *M. charantia* in ethanol could restrain the fat synthesis. The bioactive compounds in *M. charantia* can stimulate glucose transporter 4 (GLUT-4) to migrate to the cell membrane and mediate glucose uptake by increasing the adenosine monophosphate-activated protein kinase (AMPK) activity. Fatty acid oxidation, insulin tolerance, and insulin resistance were assessed by glucose tolerance test in mice. The results showed that *M. charantia* extract containing stearoyl-glycosides could increase fatty acid oxidation and glucose distribution. Thus, it can be concluded that bioactive compounds in *M. charantia* have a role in the prevention of obesity [71].

Peroxisome proliferator-activated receptor gamma (PPARγ) belongs to nuclear hormone receptor superfamily PPARs. PPARγ is overexpressed in adipocytes, vascular smooth muscle cells, macrophages, cardiomyocytes, and endothelial cells [93,94,95]. Overexpression of PPARγ gene transforms non-adipogenic embryonic fibroblasts and myoblasts into adipocytes [96].

Roffey et al., used 3T3-L1 adipocyte model to demonstrate that even the *M. charantia*-water extract can promote the utilization of glucose and secretion of adiponectin. The combination treatment of cells with 0.2 mg/mL *M. charantia* water extract and 0.5 nM insulin was associated with significant (*p* < 0.05) increase in the glucose uptake (61%) and adiponectin secretion (75%) in comparison to those observed in the control cells [73]. Popovich et al., treated 3T3-L1 cells with methanol extract of *M. charantia*. The results suggested that the concentration of TG was slightly decreased, the release of lactate dehydrogenase was increased, and the expression of adiponectin and PPARγ was blocked in the cells. The number of G2/M phase cells was increased and that of G1 phase cells was decreased, thus, reducing the accumulation of fat in the cells [72]. This is contrary to previous reports. It is likely that *M. charantia* does not interact directly with 3T3-L1 cells, such as the thiazolidinediones reported [97,98], Alternately, it was found that lipid accumulation in 3T3-L1 cells and expression of PPARγ were reduced when AMP kinase was activated [99,100]. It is thus possible that activation of AMP kinase may also play a role in reducing lipid accumulation observed in this study.

PPAR-γ has been known to play an important role in the regulation of adipocyte differentiation and various other metabolic pathways, such as sugar and fat metabolism. The GLUT-4 dysfunction, owing to aberrations in the glucose transport mechanism, is an important cause of adipose tissue insulin resistance [101].

In a study by Shih et al., the effects of *M. charantia* on skeletal muscle GLUT-4 were observed. They found that *M. charantia* extract could increase the expression of PPARγ in adipose tissue [74].

### 3.2. Promotion of the Utilisation of Glucose

Yibchokanun et al., performed experiments on 3T3 adipocytes and revealed that *M. charantia* ethanol extract significantly increased the glucose uptake after 4 and 6 h of cell incubation as the *M. charantia* extract can promote insulin secretagogue activity [75]. He et al. performed in vitro activation experiments and showed that *M. charantia* polysaccharides increased PPARγ and PPARδ activation to 1.7 and 2.0-fold, respectively [76].

Phosphatidylinositol 3-kinase (PI3K) is a key signaling molecule in the process of insulin signal transduction. Activated PI3K can accelerate the transport of GLUT-4 and GLUT-1 to the membrane, and regulate the effect of glucose on myocytes, adipocytes, and hepatocytes, by inhibiting gluconeogenesis through inhibition of the activity of enolase pyruvate kinase. Kumar et al., treated L6 skeletal muscle cells grown in 2-deoxy-D-[1-3H] glucose, with the combination of *M. charantia* water extract and chloroform (6 μg/mL) up to 2.8, 3.6, and 3.8 times of original level of extract and showed that upregulation of the glucose uptake was approximately twice as fast as uptake of insulin and rosiglitazone [77].

AMPK promotes GLUT-4 transport from the cytoplasm to the cell membrane, thereby increasing glucose uptake and fatty acid oxidation. The enhanced expression of GLUT-4 in the central part of peripheral tissues leads to enhancement in the utilization of glucose, which is probably related to the activation of AMPK pathway. Studies by Iseli and others have found that purification of the ethanol extracts obtained from *M. charantia* cells, prior to treatment of the model cells can increase AMPK activity [102]. The activation of AMPK pathway is the initiating factor of insulin. Further, Klomann et al., found that *M. charantia* extract can increase the expression of GLUT-4 and glucose uptake [103]. Han et al., findings demonstrated that the new cucurbitane-type triterpenoids isolated from the ethanol extract of *M. charantia* enhances potential for prevention and management of obesity by improving insulin sensitivity and glucose homeostasis. Their results showed that *M. charantia* extract promotes glucose utilization by increasing glucose uptake into the skeleton [78].

AMPK activates the translocation of GLUT-4, increases muscle responses to insulin, and stimulates fatty acid oxidation. In vivo experiments by Miura et al. have shown that treating obese mice with water extract of *M. charantia* resulted in the strong activation of skeletal muscle AMPK. The water extract of *M. charantia* was fed to mice at 100 mg/kg and 20 mg/kg BW and this treatment resulted in an increase in AMPK levels by 1.6-fold [79]. Zhu et al., found that crude extract of *M. charantia* could prominently reduce fasting plasma glucose (21.40 to 12.54 mmol/L) and serum insulin (40.93 to 30.74 mIU/L) in diabetic rats [80].

The inhibition of fat synthesis and promotion of glucose utilization function of *M. charantia* is related to adiponectin, leptin, GLUT-4, AMPK, PPARγ, C/EBPα, and PI3K. By promoting their regulation, it promotes fatty acid oxidation and inhibits adipocyte differentiation to achieve anti-obesity effect (Figure 4).

### 3.3. Auxiliary Lipid-Lowering Activity

Hyperlipidemia refers to the phenomenon of hypercholesterolemia and high TG levels in the blood, or low levels of HDL-c. Hyperlipidemia is one of the metabolic characteristics of obesity.

Hussain et al., carried out in vivo biological investigations with an ethanolic extract of *M. charantia*, to evaluate both its anti-hyperlipidemic activity and serum uric acid level-reducing potential. Results showed that ethanolic extract of *M. charantia*, at a dose of 200 mg/kg BW, significantly (*p* < 0.05) reduced cholesterol and TG levels when co-administrated with 20% fat and normal feed, which was comparable with data obtained from standard Atorvastatin treatment [81]. Studies by Xu and others have shown that *M. charantia* fruit powder prevents high-fat diet-induced metabolic disorders in mice. Their results also showed that both water-soluble and insoluble parts of *M. charantia* can reduce TG levels in obese mice [82].

The studies by Ahmad et al., suggested that the levels of non-esterified cholesterol and phospholipids in the serum of diabetic rats was twice as high as that of the normal control group. The TG (TG) level was four times more and the level of HDL-c was 50% more in diabetic rats than those in the control group. The crude extract of *M. charantia* treatment administered to diabetic rats could reduce the elevated levels of serum non-esterified cholesterol, phospholipids, TG, and HDL-c [83]. This effect probably occurs through the regulation of TG, cholesterol, and phospholipid metabolism in the liver and other organs, and increase in insulin secretion. Experiments by Welihinda et al., showed that the treatment of *M. charantia* fruit extract could increase the insulin secretion in the pancreas of diabetic rats, and the insulin-activated lipase hydrolyses the lipoprotein-bound TG [84]. Lipase enzyme cannot be activated in insulin-deficient individuals, so it cannot reduce TG levels. Senanayake et al., studied the effects of three different varieties (Koimidori, Powerful-Reishi, and Hyakunari) of *M. charantia* and those of methanol fraction extract of Koimidori variety on serum and liver TG in rats. Koimidori extract (3%) was found to be the most effective in reducing hepatic TG levels in experimental animals, and the effects of Hyakunari and Powerful-Reishit were successively reduced. Koimidori extracts obtained from n-hexane, acetone, and methanol, respectively, were administered to experimental animals and compared with those of Koimidori without isolation. The 1% methanol extract acts as a 3% Koimidori crude extract, concluding that the active ingredient that reduces liver TG levels in the *M. charantia* variety Koimidori is enriched in the methanol extract [85]. Studies by Jayasooriya have shown that the diet containing cholesterol tends to decrease HDL-c levels in the body, whereas consumption of *M. charantia* can cause an increase in serum HDL-c levels regardless of dietary cholesterol levels [86].

While *M. charantia* powder (organic solvent extraction) has been shown to affect serum HDL levels, but it shows only a minor effect on serum TG levels [85]. Additionally, *M. charantia* extract significantly reduces the total liver cholesterol levels and TG levels. *M. charantia* extract shows a dose-response relation while reducing the TG levels. Abd et al., found that *M. charantia* extract can significantly reduce the serum levels of cholesterol and TGs in alloxan-induced diabetic mice [87]. Ahmad et al., performed an experiment to understand the effect of the *M. charantia* fruit extract on STZ-induced diabetic rats. Elevated levels of HDL-c can result in increased lipid flow in tissues or increased free fatty acid intake and storage capacity to achieve the effect of lowering serum phospholipid and TG level [83]. Pan et al., treated hyperlipidemia model mice for 4 weeks with water extract of *M. charantia* and found that it can significantly reduce the liver fat accumulation in mice with hyperlipidemia and significantly lower the serum TC and TG concentrations [88]. Studies by Chanda and others have shown that among various parts of *M. charantia*, whole fruit part in a dose of 300 mg/kg body weight found to be suitable in reduction in cholesterol, increase in HDL and minimize the amount of LDL and TG [104].

Thus, these studies demonstrate that the effects of *M. charantia* on the liver and its serum auxiliary lipid-lowering activity may be mediated by the regulation of TG, cholesterol, and phospholipid metabolism in the liver and other organs, and increase in insulin secretion (Figure 5). However, in the previous studies, *M. charantia* showed a negligible effect on serum TG levels. The reason for this phenomenon may be due to the variations in experimental conditions.

## 4. Risk of Over-Consumption of *M. charantia*

Since ages, *M. charantia* has been widely recognized and used as medicinal edible plant. However, only a few studies have shown safety issues associated with excessive consumption of *M. charantia* [105,106,107]. Leung et al. and Raman et al. reviewed the possible side effects caused by excessive consuming *M. charantia* juice, including children consuming large amounts of *M. charantia* tea after a period of hypoglycemic coma and convulsions. They reviewed that long-term consumption of *M. charantia* will lead to the glucose-6-phosphate dehydrogenase deficiency and risk of fava bean disease in humans. The toxic lectin in the rind and pulp of *M. charantia* may inhibit the synthesis of proteins in the intestine and this effect may be related to the separation of *Vicia faba* glycosides present in the *M. charantia* seeds [108,109]. Nadkarni et al., measured the blood pressure, blood glucose, liver and kidney function, and other related indicators, after the consumption of 500 mL of *M. charantia* juice concentrate, and found that *M. charantia* juice can cause an acute gastric ulcer and intestinal bleeding. They speculated that the toxic substances producing these effects might be alkaloids, lectins, or charantin [110].

## 5. Current and Future Applications of *M. charantia* in the Anti-Obesity Treatment

In order to maximize the activity of *M. charantia* components and avoid the risk of over-consumption, the development of inventive processing method is needed. In order to obtain a new sugar-free *M. charantia* yoghurt, Zhang et al. mixed *M. charantia* juice and fresh milk as the raw materials at a 3:7 ratio along with *Lactobacillus bulgaricus* and *Streptococcus thermophilus* at a 1:1 ratio as the starter inoculum for fermentation. This yogurt with *M. charantia* fragrance was much sweeter and sourer than others. This product exhibits a high promotional value since both *M. charantia* and yoghurt possess numerous benefits to public health [111]. Flour containing 4% *M. charantia* powder was used to form a new table staple food-balsam health noodles [112]. In the fat-lean ratio was 1:4, *M. charantia* mass fraction was 9%, starch mass fraction was 12%, compound sausage with *M. charantia* not only reduced the production costs, but also improved the nutritional value [113]. *M. charantia*, peanuts and xylitol-based raw materials are used to prepare low-sugar health drinks [114]. A healthy drink is produced by mixing 30% *M. charantia* juice and 15% Campanulaceae extract added with 8% sugar and 0.1% citric acid [115]. *M. charantia* as raw material, zinc gluconate as color protector, and β-cyclodextrin as bitter embedding agent are used to produce *M. charantia* granules [116]. The best formula of effervescent tablets was determined as follows: *M. charantia* powder 4.0%; sucrose 17.8%, sweet 3.6%, sodium bicarbonate 31.1%, citric acid 6.7%, and maltodextrin 35.6%. The *M. charantia* effervescent tablet drinks have good sensory quality and unique taste; its bitter taste was masked, and it is easily accepted by consumers [117]. Fresh milk and *M. charantia* juice at 8:2 ratio, added with 9% sucrose and 0.15% compound stabilizer produces a *M. charantia* milk drink [118]. A solution of *M. charantia* juice and apricot juice at 2:7 ratio, produces a compounded drink [119]. A solution of 10% *M. charantia* juice, 20% citrus juice, 7% sugar, 3% honey, and 0.2% citric acid is formulated into a citrus-*M. charantia* compound fruit and vegetable drink [120].

Meanwhile, there are few food product using bioactive components of *M. charantia* for anti-obesity. Hopefully, this review motivates to develop the related products.

## 6. Conclusions

Recently, several developments at the interface of natural product chemistry and medicine have demonstrated that various plant extracts have good anti-obesity properties and are of interest due to their low toxicity or non-toxic benefits. In spite of its regular use as medicinal plant in many Asian countries, active ingredients of *M. charantia* have not been identified or studied systematically studied by implementing novel strategies and technologies. Although there are many studies on the mechanism of anti-adiposity at the cellular level, its mechanism of action in the animals is rarely studied and is a controversial subject. For instance, the inhibitory effect of the *M. charantia* extract on leptin expression contradicts with its anti-adiposis effect. Only a few experiments on the processing of *M. charantia* have been conducted; however, these could provide the basis for further developments. Moreover, there is no large-scale industrialized application at present and there are only a few reports on the comprehensive utilization of *M. charantia* to fully explore their potential as anti-obesity agent.

In this review, we summarized the biochemical components of *M. charantia* that show anti-obesity activity and their mechanisms of action. Additionally, we described the relevant current progress in this area. Further, we verified the function and synergistic effects of various components of *M. charantia* and provided an account on the subsequent processing methods that could lead to comprehensive utilization of *M. charantia*.

## Figures and Tables

**Figure 1 ijerph-16-03251-f001:**
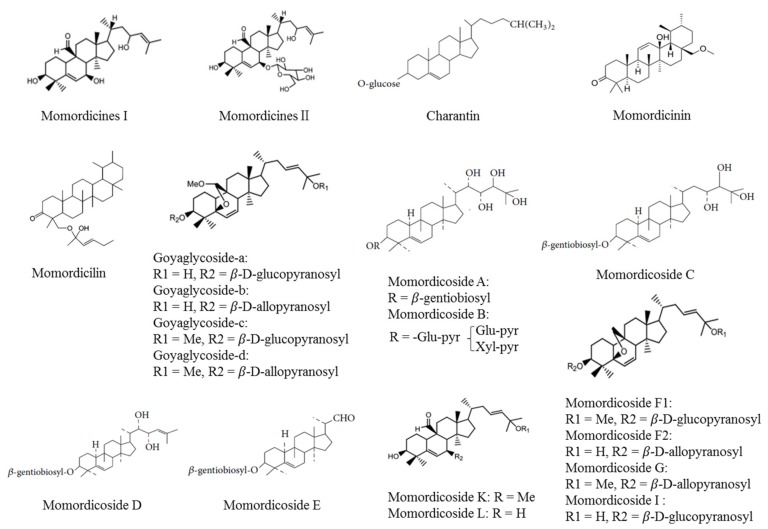
Chemical structure of saponins and triterpenes isolated from *M. charantia*.

**Figure 2 ijerph-16-03251-f002:**
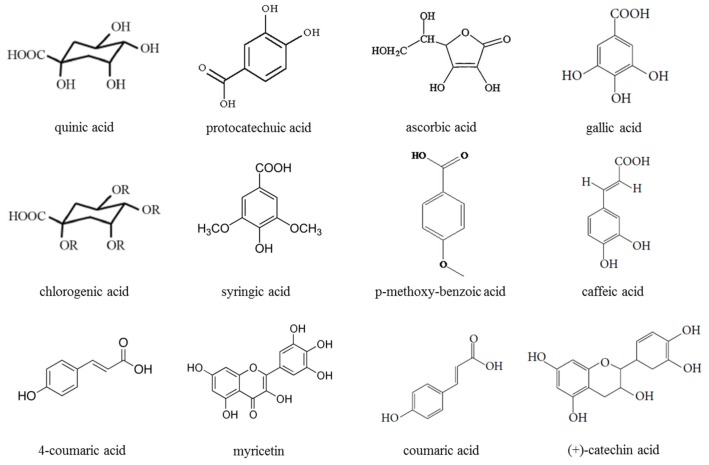
Chemical structure of phenolics and flavonoids isolated from *M. charantia*.

**Figure 3 ijerph-16-03251-f003:**
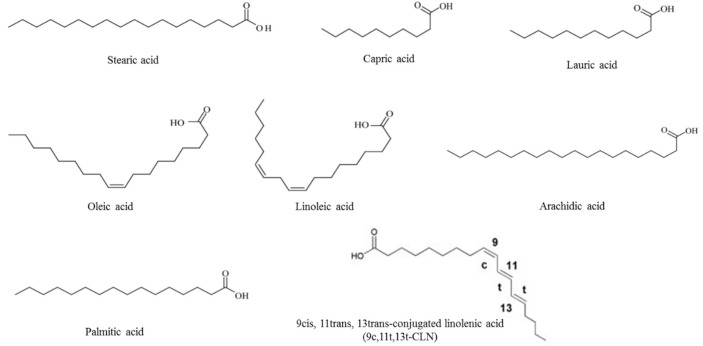
Chemical structure of fatty acids isolated from *M. charantia*.

**Figure 4 ijerph-16-03251-f004:**
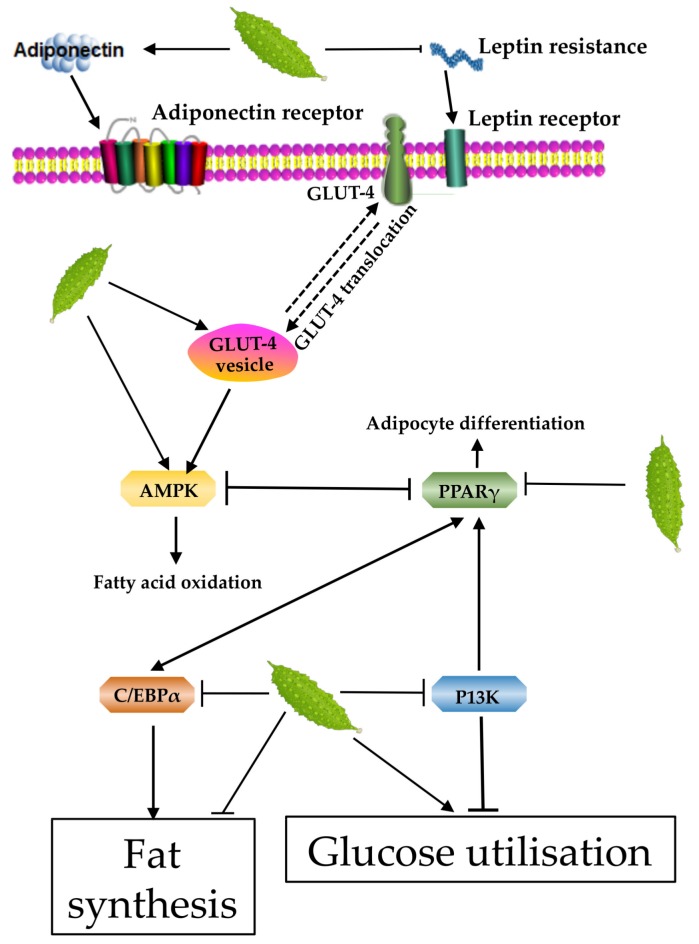
Possible mechanisms contributing to the inhibition of fat synthesis and glucose utilization of *M. charantia*. *M. charantia* stimulates glucose transporter 4 (GluT-4) to migrate to the cell membrane and mediate glucose uptake by increasing adiponectin, and the adenosine monophosphate-activated protein kinase (AMPK) activity. In addition, *M. charantia* blocks the expression of PPARγ, and PPARγ downstream pathway including C/EBPα and PI3K to reduce the accumulation of fat.

**Figure 5 ijerph-16-03251-f005:**
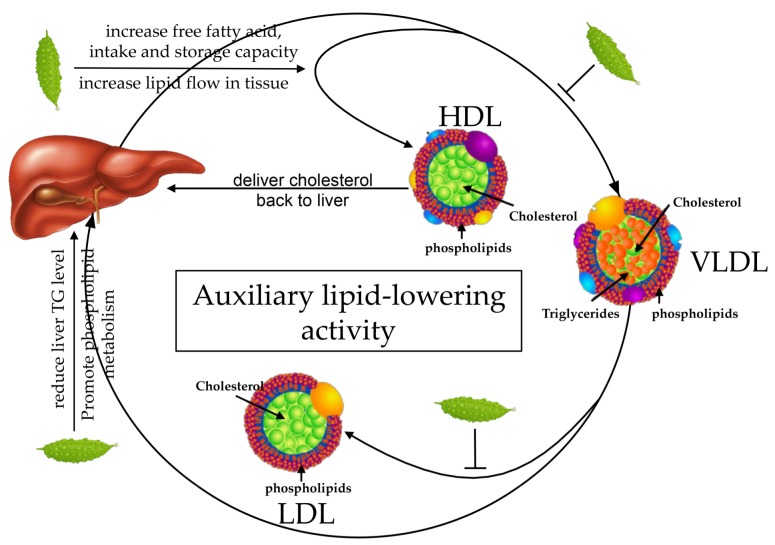
Possible mechanisms contributing to the auxiliary lipid-lowering activity of *M. charantia*. *M. charantia* reduce the elevated levels of serum cholesterol, phospholipids, triglyceride (TG), and low-density lipoprotein (LDL). On the contrary, *M. charantia* increases the high-density lipoprotein (HDL) level by elevating lipid flow in the tissues.

**Table 1 ijerph-16-03251-t001:** Anti-obesity functions of bitter gourd (*M. charantia*) crude extract.

Function	Downregulation	Upregulation	References
Inhibition of fat synthesis	pAkt, tAkt	pAMPK, tAMPK	[71]
		adiponectin	[72]
	PPARγ, adiponectin		[73]
	leptin	GLUT-4	[74]
Promotion of glucose utilization	insulin resistance		[75]
		PPARδ	[76]
		GLUT-4	[77]
		glucose utilization	[78]
		AMPK	[79]
	fasting plasma glucose		[80]
Auxiliary lipid-lowering activity	cholesterol, TG		[81]
	TG		[82]
	cholesterol, TG	HDL	[83]
	TG	insulin sensitivity	[84]
	TG	HDL	[85]
		HDL	[86]
	cholesterol, TG		[87]
	TC, TG		[88]

TG: triglyceride; TC: total cholesterol; HDL: high-density lipoprotein; AMPK: adenosine monophosphate-activated protein kinase; GLUT-4: glucose transporter 4; PPAR: Peroxisome proliferator-activated receptor; Akt: Protein kinase B.

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
