# Peer review of "The Role of Momordica charantia in Resisting Obesity"

_ijerph, 2019, doi:10.3390/ijerph16183251_

Round 1

Reviewer 1 Report

In the paper entitled “The Role of Momordica charantia in Resisting Obesity”, authors reviewed the anti-obesity effects of Momordica charantia (MC). I appreciate this work, because, despite the abundant literature regarding MC, only few reviews are focused on circumscribed topics other than anti-diabetic effects. On the other hand, the authors did not give a logical and structured organization to this review as the studies reported are often a mere list, without any critical analysis. This is fundamental, since the studies here referred, report results which are often opposite from each other. Hence, there is no clarity on the mechanism by which the MC exercises its anti-obesity effects. The sentence, reported in Conclusions section about the contradictory results obtained (line 353), should be emphasized from the beginning.

Many recent works concerning MC and obesity have not been mentioned at all (for example, PMID 31005847, 30628614, 30536964, 29892950, 29126447, 29090226, 28962621 and so on).

Although the work is quite well written, especially in the first part, many sentences are difficult to understand (for example, chapter 4, lines 308-310).

In my opinion, the work could be published on Int. J. Environ. Res. Public Health only after major revisions and after a deep reorganization of its structure.

Please, see below some suggestions that could help paper's improvement.

Chapter 2. This chapter should be better structured. The authors report several studies not concerning MC, confusing the reader. Mainly in chapter 2.2, after discussing the phenolic flavonoid compounds identified in MC, authors report studies (references 59 and 60) that do not concern this plant. I suggest focusing more on studies regarding bitter melon.

Line 73, Reference 23 does not speak about obesity at all and no studies are reported there about obesity. Regarding triterpene saponins compounds, Jager and co-workers say that “Therefore the correlation of a triterpene acid rich nutrition and the beneficial effects of consuming Mediterranean food should be investigated in more detail”. Moreover, authors of the present review should verify if it is true that, in the Mediterranean populations, obesity is absent, as they say.

Line 81, Reference 28. Authors should critically comment the finding made by Ng and co-workers; in the work cited at reference 28 and also in another one (PMID 3021185), the activity of saponins purified from M. charantia consists in the inhibition of glucagon and, in general, in anti-lipolytic and lipogenic effects. How does this relate to the anti-obesity effect? Please, explain.

Chapter 3.1, lines 181-188. The sentence “could improve fat production” (line 182) contradicts the sentence “… have a role in the prevention of obesity”. What do the authors mean?

Lines 193-197, reference 79. Were the experiments conducted with M. charantia or with M. grosvenori? How do the authors explain the correlation between the reduction of adiponectin expression with the anti-obesity effect?

Reference 80: As authors state in the text, and as reported in the original paper (cit. “The results of the present study demonstrate that a 30 min pre-treatment with the water soluble MC extract increases the glucose uptake activity at sub-optimal levels of insulin and increases adiponectin secretion of 3T3-L1 adipocytes.”), MC increase adiponectin secretion in adipocytes. Why in Table 1 is reported a downregulation? what possible explanation can be given to justify the opposite effect reported in reference 79?

I reiterate what was said earlier, also for reference 82. How do the authors explain the discrepancy between the reduction of leptin / PPARgamma and the anti-obesity effect?

Figure 4 is quite misleading as it appears that MC activates adiponectin and inhibits PI3K (whereas in the legend the contrary is stated). Please make it clearer.

Chapter 4. Sentence in lines 308-310 does not make sense. Also sentence in lines 325-326 is quite difficult to understand.

Author Response

In the paper entitled “The Role of Momordica charantia in Resisting Obesity”, authors reviewed the anti-obesity effects of Momordica charantia (MC). I appreciate this work, because, despite the abundant literature regarding MC, only few reviews are focused on circumscribed topics other than anti-diabetic effects. On the other hand, the authors did not give a logical and structured organization to this review as the studies reported are often a mere list, without any critical analysis. This is fundamental, since the studies here referred, report results which are often opposite from each other. Hence, there is no clarity on the mechanism by which the MC exercises its anti-obesity effects.

→ Thank you for your time to review our manuscript. We are very much honored it and have learned a lot from your comments. We thoroughly revised whole manuscript as your comments, and highlighted them by colored text. In addition, we also corrected typos and grammatical mistakes while we were revising the manuscript.

Point 1: The sentence, reported in Conclusions section about the contradictory results obtained (line 353), should be emphasized from the beginning.

Response 1: Thank you very much for your considerable comment. We emphasized the part in the introduction as your comment.

Point 2: Many recent works concerning MC and obesity have not been mentioned at all (for example, PMID 31005847, 30628614, 30536964, 29892950, 29126447, 29090226, 28962621 and so on).

Response 2: Thank you very much for your considerable comment. We added the part and references in the article as your comment.

Point 3: Although the work is quite well written, especially in the first part, many sentences are difficult to understand (for example, chapter 4, lines 308-310). In my opinion, the work could be published on Int. J. Environ. Res. Public Health only after major revisions and after a deep reorganization of its structure. Please, see below some suggestions that could help paper's improvement.

Response 3: Thank you very much for your considerable comment. We revised whole manuscript and point-by-point reply to your comments was described as follows. In addition, the manuscript was edited by English native speakers, and we just attached the certification at the end of this response.

Point 4: Chapter 2. This chapter should be better structured. The authors report several studies not concerning MC, confusing the reader. Mainly in chapter 2.2, after discussing the phenolic flavonoid compounds identified in MC, authors report studies (references 59 and 60) that do not concern this plant. I suggest focusing more on studies regarding bitter melon.

Response 4: Thank you very much for your considerable comment. We agree with you, and really tried to add M. charantia-related researches and references in the chapter 2.2 as your comment. As you see, we added several researches regarding M. charantia for saponins, triterpenes, and unsaturated fatty acids. But there are few clear data on the anti-obesity activity of phenolics and flavonoids in M. charantia. We reported two studies about anti-obesity activity of flavonoids that do not concern M. charantia. Even there are few data regarding M. charantia for phenolics and flavonoids, it possesses lots of phenolics and flavonoids. Therefore, we would like to provide its possibility for the future research of phenolics and flavonoids in M. charantia.

Point 5: Line 73, Reference 23 does not speak about obesity at all and no studies are reported there about obesity. Regarding triterpene saponins compounds, Jager and co-workers say that “Therefore the correlation of a triterpene acid rich nutrition and the beneficial effects of consuming Mediterranean food should be investigated in more detail”. Moreover, authors of the present review should verify if it is true that, in the Mediterranean populations, obesity is absent, as they say.

Response 5: Thank you very much for your considerable comment. We wanted to show that the Mediterranean diet (MD) has been identified as having proved to be the most effective amongst many others in terms of prevention of obesity-related diseases [1]. MD is characterized by a high intake of vegetables, fruits, nuts, cereals, whole grains, and olive oil, as well as a moderate consumption of fish and poultry, and a low intake of sweets, red meat and dairy products [2,3]. Also, many flavonoids are found in vegetables and fruits [4]. However, we agree with you. The statement would be controversial, and also that is not particularly relevance to the topic of this article. For this point, we decided to delete this part. Hope you satisfy with our decision.

Romagnolo, D.F.; Selmin, O.I. Mediterranean Diet and Prevention of Chronic Diseases. Nutr. Today 201752, 208–222. Bach-Faig, A.; Berry, E.M.; Lairon, D.; Reguant, J.; Trichopoulou, A.; Dernini, S.; Medina, F.X.; Battino, M.; Belahsen, R.; Miranda, G.; et al. Mediterranean Diet Foundation Expert Group. Mediterranean diet pyramid today. Science and cultural updates. Public Health Nutr. 201114, 2274–2284. Willett, W.C.; Sacks, F.; Trichopoulou, A.; Drescher, G.; Ferro-Luzzi, A.; Helsing, E.; Trichopoulos, D. Mediterranean diet pyramid: A cultural model for healthy eating. Am. J. Clin. Nutr. 199561, 1402S–1406S. Saxena, M., Saxena, J., Nema, R., Singh, D., & Gupta, A. (2013). Phytochemistry of medicinal plants. J. Pharmacogn. Phytochem. 1(6).

Point 6: Line 81, Reference 28. Authors should critically comment the finding made by Ng and co-workers; in the work cited at reference 28 and also in another one (c), the activity of saponins purified from M. charantia consists in the inhibition of glucagon and, in general, in anti-lipolytic and lipogenic effects. How does this relate to the anti-obesity effect? Please, explain.

Response 6: Thank you very much for your considerable comment. We wanted to speak that saponins which have insulin-like bioactivity from M. charantia ameliorate obesity by improving insulin sensitivity and glucose homeostasis. We revised the sentence to be clear as your comment.

Point 7: Chapter 3.1, lines 181-188. The sentence “could improve fat production” (line 182) contradicts the sentence “… have a role in the prevention of obesity”. What do the authors mean?

Response 7: Thank you very much for your considerable comment. We wanted to report that M. charantia could restrain the generation of adipose and ameliorate fat undesirable effect. We revised the sentence to be clearer as your comment.

Point 8: Lines 193-197, reference 79. Were the experiments conducted with M. charantia or with M. grosvenori? How do the authors explain the correlation between the reduction of adiponectin expression with the anti-obesity effect?

Response 8: Thank you very much for your considerable comment. We agree with you, we are sorry, in this part is not clear. The experiments were conducted with M. charantia.

There was a decrease in adiponectin in this report, contrary to other reports. It is likely that M. charantia does not interact directly with 3T3-L1 cells, such as the thiazolidinediones reported [1, 2]. Alternately, it was found that lipid accumulation in 3T3-L1 cells and expression of PPARγ were reduced when AMP kinase was activated [3, 4]. Therefore, it is possible that activation of AMP kinase may also play a role in reducing lipid accumulation observed in this study. We added detailed explain in the manuscript as your comment. Once again, appreciate your valuable comment.

Arner, P. The adipocyte in insulin resistance: key molecules and the impact of the thiazolidinediones. Trends Endocrinol Metab 2003, 14 (3), 137-145. Fürnsinn, C.; Waldhäusl, W. Thiazolidinediones: metabolic actions in vitro. Diabetologia 2002, 45 (9), 1211-1223. Hwang, J.-T.; Kwon, D. Y.; Yoon, S. H. AMP-activated protein kinase: a potential target for the diseases prevention by natural occurring polyphenols. N Biotechnol 2009, 26 (1-2), 17-22. Hwang, J. T.; Lee, M. S.; Kim, H. J.; Sung, M. J.; Kim, H. Y.; Kim, M. S.; Kwon, D. Y. Antiobesity effect of ginsenoside Rg3 involves the AMPK and PPAR‐γ signal pathways. Phytotherapy Research: An International Journal Devoted to Pharmacological and Toxicological Evaluation of Natural Product Derivatives 2009, 23 (2), 262-266.

Point 9: Reference 80: As authors state in the text, and as reported in the original paper (cit. “The results of the present study demonstrate that a 30 min pre-treatment with the water soluble MC extract increases the glucose uptake activity at sub-optimal levels of insulin and increases adiponectin secretion of 3T3-L1 adipocytes.”), MC increase adiponectin secretion in adipocytes. Why in Table 1 is reported a downregulation? what possible explanation can be given to justify the opposite effect reported in reference 79?

Response 9: Thank you very much for your considerable comment. We hardly tried to give a possible explanation in the opposite results. We would appreciate it if you would take the time to read it, and hope you satisfy our explanation.

Point 10: I reiterate what was said earlier, also for reference 82. How do the authors explain the discrepancy between the reduction of leptin / PPARgamma and the anti-obesity effect?

Response 10: Thank you very much for your considerable comment. As you mentioned, there are different results leptin/PPARgamma on M. charantia. In this review, we wanted and focused to prove that M. charantia extract could decrease the expression of PPARγ in adipose tissue. PPARγ, which is mainly present in adipose tissue, the intestine and macrophages, is well known for its role in regulating adipogenesis, insulin sensitivity, and inflammation [1]. Increasing evidence indicates that PPARγ is involved in the etiology and pathogenesis of numerous conditions associated with metabolic syndrome, including obesity, NAFLD, and cardiovascular disease [2], [3], making this receptor an attractive pharmacological target for the treatment and prevention of the above-mentioned metabolic disorders [4]. Meanwhile, many anti-diabetic drugs have been withdrawn due to their serious side effects including obesity. However, M. charnatia possesses both anti-diabetic and anti-obesity activities. Also, there are discrepancy among some biomarkers as your comments, So, we assume that there is another key biomarker(s) for its anti-obesity activity. We wish this review motivate the further investigate to leaders to prove it.

Auwerx, PPARgamma, the ultimate thrifty gene, Diabetologia 1999, 42, 1033–1049. E. Soccio, E.R. Chen, S.R. Rajapurkar, P. Safabakhsh, J.M. Marinis, J.R. Dispirito, M.J. Emmett, E.R. Briggs, B. Fang, L.J. Everett, H.W. Lim, K.J. Won, D.J. Steger, Y. Wu, M. Civelek, B.F. Voight, M.A. Lazar, Genetic variation determines PPARgamma function and anti-diabetic drug response in vivo, Cell 2015, 162, 33–44. Zhou, M. Febbraio, T. Wada, Y. Zhai, R. Kuruba, J. He, J.H. Lee, S. Khadem, S. Ren, S. Li, R.L. Silverstein, W. Xie, Hepatic fatty acid transporter Cd36 is a common target of LXR, PXR, and PPARgamma in promoting steatosis, Gastroenterology 2008, 134, 556–567. Wang, B. Waltenberger, E.M. Pferschy-Wenzig, M. Blunder, X. Liu, C. Malainer, T. Blazevic, S. Schwaiger, J.M. Rollinger, E.H. Heiss, D. Schuster, B. Kopp, R. Bauer, H. Stuppner, V.M. Dirsch, A.G. Atanasov, Natural product agonists of peroxisome proliferator-activated receptor gamma (PPARgamma): a review, Biochem. Pharmacol. 2014, 92, 73–89.

Point 11: Figure 4 is quite misleading as it appears that MC activates adiponectin and inhibits PI3K (whereas in the legend the contrary is stated). Please make it clearer.

Response 11: Thank you very much for your considerable comment. We revised the legend as your comment.

Point 12: Chapter 4. Sentence in lines 308-310 does not make sense. Also sentence in lines 325-326 is quite difficult to understand.

Response 12: Thank you very much for your considerable comment. We modified chapter 4 section as your comment. Once again, appreciate your valuable comment.

Reviewer 2 Report

Dear authors,

It was my pleasure to review your manuscript submitted to the Journal of Environmental Research and Public Health. Below you will find the list of my suggestions for its further improving.

The Chapter 4 (Prospects of the Development…) is critical for appreciating the nutritional potential of M.charantia and is of high interest for nutritional scientists, especially those who is interested in effects of bioactive foods. It requires significant revision. First, its first paragraph contains a lot of information not relevant to the Prospects topic. It should be better moved to the previous chapters. Next, the chapter contains several examples of foods currently proposed to customers (second paragraph), but it also does not deal much with Future Prospects. I propose to change the Chapter 4 title (for instance: Current and Future Applications…) and complement it with author’s suggestions regarding the most promising approaches to using the bioactive components of M.charantia in food industry. This analysis may be based on biochemical nature and biologic function of reviewed M.charantia bioactive components, their expected beneficial anti-obesity effects on human health and the examples of using these and/or similar natural compounds in currently available popular food products. In a Chapter 2, Anti-Obesity components of M.charantia, for all types of described bioactives compounds, the presentation of studies under review needs to be better structured. It will make the text more reader-friendly if you re-organize the data. First, tell the readers, what compounds were obtained from M.chrantia, then present subsequently the data about their use in in vitro studies, animal studies and clinical studies. The text is abundant with annoying typos and, probably, needs to be double checked by professional editor. Page 2, lines 52-54: “In traditional 53 Chinese treatments, M. charantia fruit, seeds, stem, and leaves are used as medicine for intestinal diseases, hypoglycaemia, cancer, and other viral infections. Are all listed diseases viral infections? Page 2, line3 73: “…absence of obesity in the Mediterranean populations…”. Is obesity really absent in a Mediterranean area? Page 2, lines 75-76: “…an extract from the Japanese horse chestnut Saponin mixture…” doesn’t make sense. Is there a chance that the mixture contained other bioactive compounds but for Saponins? Page 4, line 123: “…administered a dose of polyphenol (100 mg/kg BW) to mice for 30 days…”. In this and other presentations of in vivo studies, the way of bioactive compound administration should be indicated. Page 4, lines 143-144: “…induction of uncoupling protein (UCP) and enhance β-oxidation 144 in mitochondria…”. In vivo or in vitro? In what tissue(s)? It relates to all other similar study presentations (for instance, lines 163-166). Page 6, line 182: “…could improve fat production”. The Improve word needs to be explained. Page 6, line 193: “Popovich et al. treated 3T3-L1 cells with methanol extract of M. grosvenori”. How is it related to M.charantia? Page 7, lines 215-217: “He et al. performed in vitro activation experiments and showed 216 that M. charantia polysaccharides PPARγ and PPARδ activation increased to 1.7 and 2.0-fold, 217 respectively”. Doesn’t make sense. Page 8, line 249: “…M. charantia blocked expression of…”. Please indicate what exactly has been used in the described study (ethanol extract etc). Page 9, lines 273-74: “…three different varieties (Koimidori, Powerful-Reishi, and Hyakunari)…”. It should be mentioned and briefly discussed in the Introduction section, that M.charantia has different varieties.

Best regards

Author Response

It was my pleasure to review your manuscript submitted to the Journal of Environmental Research and Public Health. Below you will find the list of my suggestions for its further improving.

Thank you for your valuable time to review our manuscript. We are very much honored it and learned a lot from your supportive comments and useful suggestions. We also thoroughly revised whole manuscript as your comments, and our point-by-point reply to your comments was described as follows. In addition, other minor errors that we found while revising the manuscript were also corrected.

Point 1: The Chapter 4 (Prospects of the Development…) is critical for appreciating the nutritional potential of M.charantia and is of high interest for nutritional scientists, especially those who is interested in effects of bioactive foods. It requires significant revision. First, its first paragraph contains a lot of information not relevant to the Prospects topic. It should be better moved to the previous chapters.

Response 1: Thank you very much for your considerable comment. In the process of searching for the bioactive food M.charantia, we found several rare reports of toxicity. We think this is very important and can provide some help for researchers who are interested in the bioactive foods of M.charantia. So we put these reports in front of the examples of foods currently. However, after your comment, we realized the first paragraph would be better moved to another chapter as your suggestion. We made new chapter for the issue. Hope you satisfy our revision.

Point 2: Next, the chapter contains several examples of foods currently proposed to customers (second paragraph), but it also does not deal much with Future Prospects. I propose to change the Chapter 4 title (for instance: Current and Future Applications…) and complement it with author’s suggestions regarding the most promising approaches to using the bioactive components of M.charantia in food industry. This analysis may be based on biochemical nature and biologic function of reviewed M.charantia bioactive components, their expected beneficial anti-obesity effects on human health and the examples of using these and/or similar natural compounds in currently available popular food products.

Response 2: Thank you very much for your considerable comment. We absolutely agree with you. However, there are few reports for the information of bioactive components food production of M. charantia. We assume there are few anti-obesity food products from M. charantia bioactive components. Hopefully, this review motivates to develop the related products.

Point 3: In a Chapter 2, Anti-Obesity components of M.charantia, for all types of described bioactives compounds, the presentation of studies under review needs to be better structured. It will make the text more reader-friendly if you re-organize the data. First, tell the readers, what compounds were obtained from M.chrantia, then present subsequently the data about their use in in vitro studies, animal studies and clinical studies.

Response 3: Thank you very much for your considerable comment. We absolutely agree with you and re-organized the sentences as your comment.

Point 4: The text is abundant with annoying typos and, probably, needs to be double checked by professional editor. Page 2, lines 52-54: “In traditional 53 Chinese treatments, M. charantia fruit, seeds, stem, and leaves are used as medicine for intestinal diseases, hypoglycaemia, cancer, and other viral infections”. Are all listed diseases viral infections?

Response 4: Thank you very much for your considerable comment. We apologize that it is not clear. We revised whole manuscript and attached the certification of English editing by native speaker. If you can take some time to check it again, we will very appreciate you

Point 5: Page 2, line3 73: “…absence of obesity in the Mediterranean populations…”. Is obesity really absent in a Mediterranean area?

Response 5: Thank you very much for your considerable comment. We wanted to speak that the Mediterranean Diet (MD) has been identified as having proved to be the most effective amongst many others in terms of prevention of obesity-related diseases [1]. MD is characterized by a high intake of vegetables, fruits, nuts, cereals, whole grains, and olive oil, as well as a moderate consumption of fish and poultry, and a low intake of sweets, red meat and dairy products [2,3]. Also, many flavonoids are found in vegetables and fruits [4]. However, we agree with you. The statement would be controversial, and that is not particularly relevance to the topic of this article. For this point, we decided to delete this part. Hope you satisfy with our decision.

Romagnolo, D.F.; Selmin, O.I. Mediterranean Diet and Prevention of Chronic Diseases. Nutr. Today 201752, 208–222. Bach-Faig, A.; Berry, E.M.; Lairon, D.; Reguant, J.; Trichopoulou, A.; Dernini, S.; Medina, F.X.; Battino, M.; Belahsen, R.; Miranda, G.; et al. Mediterranean Diet Foundation Expert Group. Mediterranean diet pyramid today. Science and cultural updates. Public Health Nutr. 201114, 2274–2284. Willett, W.C.; Sacks, F.; Trichopoulou, A.; Drescher, G.; Ferro-Luzzi, A.; Helsing, E.; Trichopoulos, D. Mediterranean diet pyramid: A cultural model for healthy eating. Am. J. Clin. Nutr. 199561, 1402S–1406S. Saxena, M., Saxena, J., Nema, R., Singh, D., & Gupta, A. (2013). Phytochemistry of medicinal plants. J. Pharmacogn. Phytochem. 1(6).

Point 6: Page 2, lines 75-76: “…an extract from the Japanese horse chestnut Saponin mixture…” doesn’t make sense. Is there a chance that the mixture contained other bioactive compounds but for Saponins?

Response 6: Thank you very much for your considerable comment. After your comment, we realized the the statement would be controversial, so decided to delete this part. Hope you satisfy with our decision.

Point 7: Page 4, line 123: “…administered a dose of polyphenol (100 mg/kg BW) to mice for 30 days…”. In this and other presentations of in vivo studies, the way of bioactive compound administration should be indicated.

Response 7: Thank you very much for your considerable comment. We tried to add the amount of bioactive compound administration as your comment.

Point 8: Page 4, lines 143-144: “…induction of uncoupling protein (UCP) and enhance β-oxidation 144 in mitochondria…”. In vivo or in vitro? In what tissue(s)? It relates to all other similar study presentations (for instance, lines 163-166).

Response 8: Thank you very much for your considerable comment. We tried to describe in detail as your comment. We wanted to speak M. charantia contains abundant unsaturated fatty acids, and the fatty acids are very benefit for health. Hopefully you satisfy our response.

Point 9: Page 6, line 182: “…could improve fat production”. The Improve word needs to be explained.

Response 9: Thank you very much for your considerable comment. We revised the sentence to be clearer as your comment.

Point 10: Page 6, line 193: “Popovich et al. treated 3T3-L1 cells with methanol extract of M. grosvenori”. How is it related to M.charantia?

Response 10: Thank you very much for your considerable comment. We are sorry, in this part is not clear. The experiments were conducted with M. charantia.

There was a decrease in adiponectin in this report, contrary to other reports. It is likely that M. charantia does not interact directly with 3T3-L1 cells, such as the thiazolidinediones reported [1, 2]. Alternately, it was found that lipid accumulation in 3T3-L1 cells and expression of PPARγ were reduced when AMP kinase was activated [3, 4]. Therefore, it is possible that activation of AMP kinase may also play a role in reducing lipid accumulation observed in this study. We added detailed explain in the manuscript as your comment. Once again, appreciate your valuable comment.

Arner, P. The adipocyte in insulin resistance: key molecules and the impact of the thiazolidinediones. Trends Endocrinol Metab 2003, 14 (3), 137-145. Fürnsinn, C.; Waldhäusl, W. Thiazolidinediones: metabolic actions in vitro. Diabetologia 2002, 45 (9), 1211-1223. Hwang, J.-T.; Kwon, D. Y.; Yoon, S. H. AMP-activated protein kinase: a potential target for the diseases prevention by natural occurring polyphenols. N Biotechnol 2009, 26 (1-2), 17-22. Hwang, J. T.; Lee, M. S.; Kim, H. J.; Sung, M. J.; Kim, H. Y.; Kim, M. S.; Kwon, D. Y. Antiobesity effect of ginsenoside Rg3 involves the AMPK and PPAR‐γ signal pathways. Phytotherapy Research: An International Journal Devoted to Pharmacological and Toxicological Evaluation of Natural Product Derivatives 2009, 23 (2), 262-266.

Point 11: Page 7, lines 215-217: “He et al. performed in vitro activation experiments and showed 216 that M. charantia polysaccharides PPARγ and PPARδ activation increased to 1.7 and 2.0-fold, 217 respectively”. Doesn’t make sense.

Response 11: Thank you very much for your considerable comment. We revised the sentence as your comment.

Point 12: Page 8, line 249: “…M. charantia blocked expression of…”. Please indicate what exactly has been used in the described study (ethanol extract etc).

Response 12: Thank you very much for your considerable comment. We made the figure based on several cited references and the function was summarized in Table 1. Regarding adiponectin, there was an opposite result with others, and we discussed about them in the chapter 3.

Point 13: Page 9, lines 273-74: “…three different varieties (Koimidori, Powerful-Reishi, and Hyakunari)…”. It should be mentioned and briefly discussed in the Introduction section, that M.charantia has different varieties.

Response 13: Thank you very much for your considerable comment. We added the part and references as well in the introduction as your comment. Once again, appreciate your valuable comment.

Round 2

Reviewer 1 Report

I appreciated the efforts made by the Authors to improve the paper that is now better focused and balanced. In my opinion, the work can be published on IJERPH in present form.